# A Case of Vancomycin-Induced Severe Immune Thrombocytopenia

**Shivani Shah** [1,*], **Ryan Sweeney** [1], **Maitreyee Rai** [2] and **Deep Shah** [2]

1   Department of Internal Medicine, Allegheny Health Network, Pittsburgh, PA 15212, USA
2   Division of Hematology and Cellular Therapy, Allegheny Health Network, Pittsburgh, PA 15212, USA
*   Correspondence: shivani.shah@ahn.org; Tel.: +1-440-334-0890

**Abstract:** A male in his 60s presented with left lower extremity fractures following a vehicle accident. Hemoglobin, initially, was 12.4 mmol/L, and platelet count was 235 k/mcl. On day 11 of admission, his platelet count initially dropped to 99 k/mcl, and after recovery it rapidly decreased to 11 k/mcl on day 16 when the INR was 1.3 and aPTT was 32 s, and he continued to have a stable anemia throughout admission. There was no response in platelet count post-transfusion of four units of platelets. Hematology initially evaluated the patient for disseminated intravascular coagulation, heparin-induced thrombocytopenia (anti-PF4 antibody was 0.19), and thrombotic thrombocytopenic purpura (PLASMIC score of 4). Vancomycin was administered on days 1–7 for broad spectrum antimicrobial coverage and day 10, again, for concerns of sepsis. Given the temporal association of thrombocytopenia and vancomycin administration, a diagnosis of vancomycin-induced immune thrombocytopenia was established. Vancomycin was discontinued, and 2 doses of 1000 mg/kg of intravenous immunoglobulin 24 h apart were administered with the subsequent resolution of thrombocytopenia.

**Keywords:** immune thrombocytopenia; drug-induced thrombocytopenia; vancomycin





## 1. Background

Immune thrombocytopenia (ITP) is a rare etiology of thrombocytopenia, with drug-induced immune thrombocytopenia (DITP) being even less common. Though the phenomena of drug-induced thrombocytopenia may be well studied, it is one that rarely presents in real life scenarios. In the United States, ITP has an incidence of approximately 12/100,000 in adults; however, drug-induced immune thrombocytopenia (DITP) is even less commonly reported [1]. Here, we present a case of vancomycin-induced immune thrombocytopenia, which highlights the importance of prompt recognition and the need for the discontinuation of the offending agent, as well as the initiation of therapeutic interventions including any or all of platelet transfusions (if there are concerns for active bleeding), intravenous immune globulin (IVIG), and glucocorticoids.

## 2. Case Presentation

A male in his 60s with a past medical history of alcohol use presented after a motor vehicle accident. Initial labs were pertinent for a hemoglobin of 12.4 mmol/L and a platelet count of 235 k/mcl (Table 1).

He received 4 units of whole blood and desmopressin in the emergency department and underwent the repair of multiple left lower extremity fractures on day 1. He was given broad spectrum antibiotics: vancomycin and cefepime from days 1 to 7, along with subcutaneous heparin for deep vein thrombosis prophylaxis. Platelets were initially stable post-operatively; however, the first drop in platelets was seen post-operatively on day 2 to 99 k/mcl, which was shortly after the start of vancomycin. Initially, it was presumed that the thrombocytopenia was potentially secondary to the patient's surgery and/or

due to a sepsis-like picture; therefore, antibiotics were continued. The hospital course was complicated by the below-knee amputation for the non-viable left foot on day 5, Staphylococcus epidermidis bacteremia on day 10, *Acinetobacter baumannii* pneumonia on day 14, right pleural effusion requiring chest tube placement, and an acute kidney injury requiring continuous renal replacement (CRRT) therapy. However, the platelet count had started to improve during this time. While CRRT was an additional potential cause of thrombocytopenia that was given consideration along with post-transfusion purpura, worsening thrombocytopenia was once again noted when vancomycin was reinitiated on days 10–17 for staphylococcus epidermidis bacteremia as the platelets dropped on hospital day 11 to 132 k/mcl immediately after the re-initiation of vancomycin. The rapid drop in platelets continued to 11 k/mcl on day 16. During this time, the patient received four units of platelets with no improvement in platelet count, and hematology was consulted for transfusion–refractory thrombocytopenia.

**Table 1.** Patient's laboratory work on day 1 of admission.

| Lab Value | Patient's Lab Value | Normal Range |
| --- | --- | --- |
| Hemoglobin | 12.4 g/dL L | 14.0–17.4 g/dL |
| White blood cell count (WBC) | 22.75 k/mcL H | 4.4–11.0 k/mcL |
| Platelet count | 235 k/mcL | 145–445 k/mcL |
| Mean corpuscular volume (MCV) | 96.9 fL | 80–96.0 fL |
| aPTT | 25 s | 23–34 s |
| PT | 14.8 s H | 11.8–14.3 s |
| INR | 1.2 H | 0.9–1.1 |
| Sodium | 131 mmol/L L | 136–145 mmol/L |
| Potassium | 3.5 mmol/L | 3.5–5.2 mmol/L |
| Chloride | 98 mmol/L | 98–107 mmol/L |
| $CO_2$ | 17 mmol/L L | 22–30 mmol/L |
| BUN | 20 mg/dL | 8–23 mg/dL |
| Creatinine | 1.54 mg/dL H | 0.7–1.2 mg/dL |
| AST | 396 U/L H | 0–40 U/L |
| ALT | 222 U/L H | 0–41 U/L |
| B12 | 1083 * | 232–1245 pg/mL |
| Folate | 10.8 * | >4.9 ng/mL |

* Labs not on admission but during hospital course. (PTT = partial thromboplastin time; PT = prothrombin time; INR = international normalized ratio; $CO_2$ = carbon dioxide; BUN = blood urea nitrogen; AST = aspartate aminotransferase; ALT = alanine aminotransferase; dL = deciliter; mg = milligram; pg = picograms; ng = nanogram; mmol = millimole; U =Unit; H = high; L = low.

A thorough workup performed by the hematology team showed a sodium citrate tube with a platelet count of 8 k/mcL (making pseudothrombocytopenia unlikely) and an international normalized ratio (INR) of 1.2 along with an activated partial thromboplastin time (aPTT) of 25 s. The patient had a stable anemia with Hb between 7.0 and 9.0 g/dL throughout admission. There were no signs of hemolysis with a normal haptoglobin (153 mg/mL) and normal bilirubin (total bilirubin 0.4 mg/dL). Additionally, the complete blood count with differential noted an elevated absolute neutrophil count (ANC) of 19.26 k/mcL, and polychromatic and hypochromatic red blood cells without any schistocytes were noted. A peripheral blood smear confirmed the above findings and ruled out platelet clumping. Liver function tests were non-elevated (Table 1). Although a CT of the abdomen/pelvis did show hepatic steatosis, there was no overt evidence of cirrhosis. The fibrinogen level was 617 mg/dL. The anti-PF4 antibody was 0.19. Other causes including nutritional deficiencies (B12 and folate), splenomegaly, and liver dysfunction secondary to alcohol use were ruled out (Table 1).

The differential diagnosis after the initial workup described above included drug-induced immune thrombocytopenia vs. heparin-induced thrombocytopenia (HIT) vs. thrombotic thrombocytopenic purpura (TTP) vs. immune thrombocytopenia. The 4T score for heparin-induced thrombocytopenia (HIT) was 3, and the anti-PF4 antibody was 0.19,

ruling out HIT. The PLASMIC score was 4 points; therefore, there was a low probability of thrombotic thrombocytopenic purpura in the given clinical setting with the absence of hemolysis. There was low suspicion for disseminated intravascular coagulation given a fibrinogen of 617 mg/dL and an International Society of Thrombosis and Haemotasis (ISTH) DIC score of 4 points. Ultimately, immune thrombocytopenia was suspected, likely drug-induced immune thrombocytopenia based on the tangential relationship of the two drops in platelet counts that occurred during the admission with two vancomycin exposures and, importantly, the patient's abrupt and rapid second drop in platelet count coincided with the re-introduction of vancomycin on hospital day 10 (Figure 1). Additionally, the patient's degree of thrombocytopenia was severe with a total count < 20 k/mcl which is more commonly associated with immune thrombocytopenia.

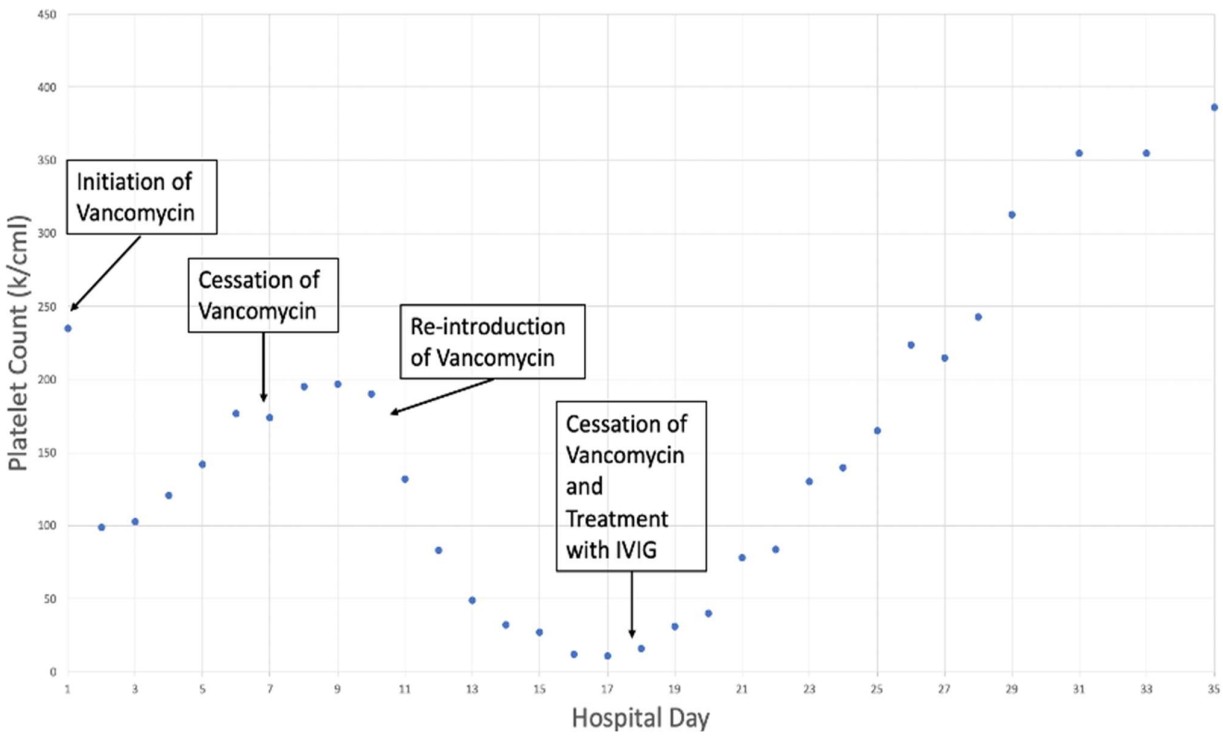

**Figure 1.** Platelet count trend upon initiation and cessation of vancomycin.

The patient was initiated on vancomycin on admission for broad spectrum antimicrobial coverage, which was subsequently discontinued on day 7. Upon the re-introduction of vancomycin on day 10, the platelet count abruptly decreased to a nadir of 11 k/mcl. The vancomycin was discontinued on day 17 with concomitant IVIG treatment and the subsequent resolution of thrombocytopenia.

The vancomycin was promptly discontinued on day 18, and 2 doses of IVIG 1000 mg/kg were given 24 h apart. Steroids were avoided in this case, due to underlying bacteremia. Platelet counts subsequently increased from 11 k/mcl to 84 k/mcl over the next 4 days, eventually recovering to 386 k/mcl by discharge. The patient was discharged with stable platelet counts and followed up with his primary care physician. Over 2 months after discharge, his platelet count remained stable at 355 k/mcl. We recommended that the patient should remain off vancomycin indefinitely.

## 3. Discussion

Immune thrombocytopenia (ITP) is caused by autoantibodies against platelets, which can be due to a primary etiology or secondary to infections, immunodeficiency syndromes, autoimmune diseases, or medications. Drugs can cause thrombocytopenia via immune or non-immune mechanisms (myelosuppression, myelotoxicity, or even megakaryocyte

dysfunction). More than 300 agents have been associated with DITP, including heparin, GPIIb/IIIa inhibitors, carbamazepine, ibuprofen, mirtazapine, oxaliplatin, quinine, quinidine, rifampicin, and antibiotics including penicillin, trimethoprim-sulfamethoxazole, ceftriaxone, and vancomycin [2,3]. Here, we presented a rare case of drug-induced thrombocytopenia secondary to vancomycin.

Vancomycin is thought to induce immune thrombocytopenia in a mechanism involving drug-dependent antibodies (DDabs), which induce platelet destruction in the reticuloendothelial system upon drug exposure [2,3]. This mechanism was first characterized with the anti-malarial agent quinine, and is thus often referred to as "quinine-type" DITP [4]. Vancomycin-induced DDabs bind to GPIIb/IIIa on the surface of platelets in a mechanism that is promoted by drug presence. A recent proposed mechanism involves the drug interaction with the complementarity-determining region (CDR) of the antibody, which enhances the affinity for the platelet glycoprotein epitope [5].

DITP is a diagnosis of exclusion that is frequently overlooked; therefore, clinical history and presentation are critical in establishing a diagnosis of this rare phenomenon. Typical presentations of DITP include signs of bleeding such as petechiae, bruising, epistaxis, GI bleeding, or intracranial hemorrhage with an accompanying acute, severe drop in platelets (commonly <20 k/mcl). Typically, thrombocytopenia occurs 5–10 days after the first drug exposure. Our patient was sensitized to vancomycin on hospital day 1 and upon re-administration developed an abrupt decrease in platelet count with a platelet nadir of 11 k/mcl.

Previously, Hackett et al. developed clinical criteria for establishing a diagnosis, including thrombocytopenia while taking the offending agent, which then resolves upon discontinuation with other causes of thrombocytopenia being excluded. Additionally, re-challenge with the offending drug demonstrates the recurrence of thrombocytopenia, or drug-dependent platelet antibodies are identified in vitro [5]. Diagnosis is established with either a positive re-challenge test or an in vitro test [2,6]. Specifically, assays can help measure the strength of the interaction between the antibodies and platelets (for example, via immunofluorescence test, immunoassays, antiglobulin based immunoassays, or flow cytometry). Flow cytometry is the most commonly used method. We did not have the option to complete an in vitro test for the detection of drug-dependent platelet autoantibodies, which could provide further confirmatory data. For this reason, we relied more heavily on clinical picture and laboratory data to arrive at the diagnosis based on the above criteria. In this case, strong evidence favoring DITP or vancomycin-induced immune thrombocytopenia (VIIT) includes the exposure to a previously identified causative drug, normalized platelet counts after the discontinuation of that drug (day 7), and re-exposure to the drug resulting in recurrent thrombocytopenia (day 10). Additionally, other causes of thrombocytopenia including heparin-induced thrombocytopenia, thrombotic thrombocytopenic purpura, splenic sequestration, nutritional deficiencies, and liver dysfunction were ruled out. When complex and severely ill patients, such as ours, are hospitalized, it is crucial to take a thorough history and note every medication that the patient has taken, including over the counter medications, herbal supplements, and quinine-containing beverages. One must also note any new medications that they have started as an inpatient. This step can easily be overlooked [7]. It can be difficult to discern which medications may have lead to the thrombocytopenia; therefore, we recommend using a database (https://www.ouhsc.edu/platelets, accessed on 3 March 2023). Using a database that reports all case reports of DITP can help determine if DITP remains a point of concern as well [8].

While thrombocytopenia is common in hospitalized patients (especially in the intensive care unit), it is often also multifactorial. This case highlights a scenario of DITP in a hospitalized patient in the setting of various other comorbidities and hospital complications and confounding factors such as sepsis, post-op thrombocytopenia, and possibly CRRT. Therefore, while the link between thrombocytopenia and a specific drug is not a straightforward diagnosis, components of the case including the severity of the thrombocytopenia

and temporal nature of the platelet fall with vancomycin led to clinical suspicion for DITP which was finally confirmed clinically with a response to treatment.

The primary treatment for DITP includes stopping the offending agent. In general, platelet counts should begin to recover within 1–2 days of stopping the offending agent and can return to a normal range within a week. Additionally, platelets are transfused to increase critically low counts, if this is for transient effect only in the case that the patient is bleeding. Glucocorticoids and IVIG can also be used in cases of severe thrombocytopenia, as often DITP and ITP are indistinguishable. Glucocorticoids lead to the apoptotic death of autoantibody-producing lymphocytes, and IVIG interferes with the macrophage consumption of autoantibody-coated platelets [9,10]. Efficacy of steroids in patients with ITP has been previously reviewed and shown that the initial response rate (>30 $\times$ 10$^7$ platelets in 7 days) is approximately 70% [11]. Three commonly used steroids include high dose dexamethasone, prednisone, and prednisolone. The initial platelet response rate based on 7 randomized controlled trials vary from 10 to 100%, 17 to 69.1 %, and 43.3 to 100% in these 3 steroids, respectively. An adequate initial platelet response is defined as a platelet count of >30 k/mcl over at least 7 days [11]. A 2016 meta-analysis included a total of 1138 patients and evaluated the efficacy of high dose glucocorticoids (3 total cycles) versus prednisone (1 mg/kg for 2–4 weeks) in patients with previously untreated ITP. Findings showed that dexamethasone leads to a more rapid increase in platelet count at 2 weeks of treatment, compared to prednisone (79% versus 59%). However, at 6 months of treatment, the platelet response showed a less notable difference (54% with dexamethasone and 43% with prednisone) [10]. Therefore, if a faster platelet response is needed, dexamethasone may be preferred. Further, there is no dose taper needed with dexamethasone, which can be a secondary advantage as well.

The response rate with the treatment of acute ITP with IVIG has been approximately 60% [12,13]. IVIG is typically administered at 1 mg/kg $\times$ 2 or 400 mg/k $\times$ 5 24 h apart and has a more transient effect (lasting approximately 3–4 weeks) but will increase the platelet count more rapidly than steroids alone (typically within 24–96 h). It is indicated in patients that may have severe thrombocytopenia or bleeding or a need for a rapid increase in platelet counts, and is typically used more as a rescue therapy in ITP as long-term toxicity can be a point of concern [14,15].

When treating ITP, important points to consider are the potential adverse effects of steroids and IVIG, which can also help discern which treatment is more appropriate. Specifically, with the long-term use of steroids, adverse effects such as hypertension, weight gain, myopathy, and mood swings should be recognized. Additionally, patients can also have an increased risk of osteoporosis and glaucoma. It is important to be wary of these side effects and weigh the risks/benefits if treating a patient with an extended course of steroids. IVIG, most notably, can lead to transfusion-related reactions, and rarely, thrombotic events [15]. Sometimes, sucrose containing IVIG products can also lead to renal impairment. More minor commonly reported side effects can include headaches, arthralgias, chills, and back pain.

Finally, a treatment option specifically in Rh-positive patients includes anti-D immune globulin. This can be used in lieu of traditional IVIG in this specific patient population. One advantage of anti-D immune globulin in comparison to IVIG includes its cheaper cost and shorter administration period. The immune globulin works against the D antigen, which ultimately prevents the binding of autoantibodies in ITP to platelets. The starting dose is 75 mg/kg of anti-D immune globulin. There are side effects to be cautious of with this treatment as well. Some overlap with IVIG with regards to side effects including fever, chills, and headaches. However, important to note are the black box warnings. These include intravascular hemolysis and multiorgan dysfunction. Patients treated with anti-D globulin should be closely monitored in the immediate post-administration period given these severe side effects [15].

Our patient was given IVIG for a rapid increase in platelet counts as his thrombocytopenia was refractory to previous transfusions. Steroids were avoided in this patient

in the setting of *Staphylococcus epidermiditis* bacteremia. After treatment with IVIG and the discontinuation of vancomycin, platelet counts promptly recovered and stabilized to 386 k/mcl by discharge on hospital day 35. Many times, the drug-dependent antibodies can be present for many years afterward; therefore, patients who have a confirmed diagnosis of DITP should avoid any further exposure to the drug, and this should be clearly indicated within the medical record to prevent future unintentional errors and potential patient morbidity.

## 4. Conclusions

Our case highlights the importance of prompt recognition and clinical criteria for establishing drug-induced immune thrombocytopenia (DITP). Thrombocytopenia in hospitalized patients is often multifactorial, and DITP can be a rather difficult diagnosis to confirm; therefore, we discussed important laboratory workup and clinical components to consider. The primary treatment for DITP remains drug discontinuation. Further, treatment options including steroids, IVIG, and anti-D globulin were also mentioned with caveats listed for a case-by-case basis to prevent severe bleeding complications in patients with severe DITP.

**Author Contributions:** For the original manuscript draft and conceptualization, both S.S. and R.S. contributed. Review and editing were completed by S.S., M.R. and D.S. Supervision was provided by D.S. All authors have read and agreed to the published version of the manuscript.

**Funding:** This research received no external funding.

**Institutional Review Board Statement:** Ethical review and IRB approval was exempted as this case study de-identified the patient's medical records. Further, there was no collection, use, or transmission of patient identifiable data. The use or disclosure of protected health information involves no more than minimal risk to the privacy of individuals; therefore, IRB review was not required.

**Informed Consent Statement:** Patient information was de-identified to protect patient health information. No consent was required or obtained.

**Data Availability Statement:** Not applicable.

**Conflicts of Interest:** The authors declare no conflict of interest.

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
