# Peer review of "A Case of Vancomycin-Induced Severe Immune Thrombocytopenia"

_hematolrep, doi:10.3390/hematolrep15020028_

Round 1

Reviewer 1 Report

Respected colleagues,

I suggest following corrections:

1. Include Keywords in Abstract

2. Include Abbreviations under Table 1

3. Improve Background, regarding explanation of relevance of your Case-report, as it is not novelty in the field

4. Include Conclusion after Discussion section

5. Include description of Methods used in laboratory diagnostic

Author Response

  1. Keywords have been added.

  1. Abbreviations under Table 1 have been added

  1. Improve Background, regarding explanation of relevance of your Case-report, as it is not novelty in the field

Though thrombocytopenia post administration of antibiotics is a well known phenomena, it still remains exceedingly rare in presentation. We aim to highlight this rare entity and hope that it will encourage physicians to continue to include this in their differential, allowing for prompt recognition and treatment, if indicated. Please see noted changes in the introduction.

  1. Include Conclusion after Discussion section:

This has been added to include the major take aways of our case report and the highlighting points.

  1. Include description of Methods used in laboratory diagnostic:

As this is a case report, I did not believe that a Methods section would be necessary. I did additionally look through prior case reports published in Hematology Reports and did not find a Methods sections in those cases. Please do let me know if there is a component that I may not be understanding. Thank you.

Reviewer 2 Report

The authors describe a case of drug-induced thrombocytopenia due to vancomycin in a patient in an elderly patient with multimorbidity.

Vancomycin as a cause of immune-mediated thrombocytopenia is rare but well known. Moreover, the clinical history of the patient poses, particularly in the first phase, more possible mechanisms possibly causing a transient thrombocytopenia. The authors mentioned them briefly and concentrated themselves of the diagnosis of drug-induced thrombocytopenia, which has apparently done only as an exclusion one.

Major comments

To make the case report more interesting for the readers, the possible causes of thrombocytopenia in the first phase of the clinical history (e.g. postoperative period, sepsis) should be better described and discussed in light of a possible masking of a shortly later occurred thrombocytopenia suggestive drug-induced thrombocytopenia. This would give the report a probably worth to be published didactic value.

Did the author perform an in vitro test using control platelets with the patient's serum in the presence and in the absence of the drug?  This would confirm their suspicion and would add interest to the case report. 

Author Response

  1. To make the case report more interesting for the readers, the possible causes of thrombocytopenia in the first phase of the clinical history (e.g. postoperative period, sepsis) should be better described and discussed in light of a possible masking of a shortly later occurred thrombocytopenia suggestive drug-induced thrombocytopenia. This would give the report a probably worth to be published didactic value.

Thank you for your comments. I agree that including differential diagnoses in the initial hospital course may be of worth. I included some discussion as how sepsis, post operative period, CRRT could all be confounding factors, as discussed later on. The temporal nature of the thrombocytopenia and then resolution with discontinuation is the offending agent is what lead to more of a DITP picture and also we thought of it more as a diagnosis of exclusion.

  1. Did the author perform an in vitro test using control platelets with the patient's serum in the presence and in the absence of the drug?  This would confirm their suspicion and would add interest to the case report. 

This is a wonderful thought, and one that was considered; however, our institution did not provide this as an option. Therefore, it was not performed. I agree; however, that this would have indeed confirmed our clinical suspicion.

Reviewer 3 Report

This is indeed an interesting case of DITP, which has been presented nicely and provides information about the patients progress through the course of admission as well as competing diagnosis; however, thrombocytopenia is a rather well-known side effect of antibiotics including vancomycin, and therefore the novelty is lacking in this report. Besides, as the authors have stated, there are multiple confounding factors throughout the course of admission, such as bacteremia and surgeries/procedures such as CRRT, limb amputation, etc all of which could have contributed (at least to some degree) to the thrombocytopenia, and it would be impossible to discern that DITP was the sole cause of thrombocytopenia in this context.  

Author Response

I agree, that there are indeed confounding factors that may have contributed to some degree of thrombocytopenia; however, the temporal nature of the thrombocytopenia with administration of vancomycin and then subsequent stopping of the offending agent lead more to a DITP picture and additionally, it was more of a diagnosis of exclusion.  

Thrombocytopenia is a well- known side effect of antibiotics; however, it is rarely diagnosed, with an incidence of about 10 persons per million are affected annually, demonstrating it’s rare diagnosis (1). Further, we would like to demonstrate the importance of keeping this medical complication on the differential diagnosis, despite it’s rare occurrence as it can resolve if treated appropriately as demonstrated in our case.

  1. Aster RH, Curtis BR, McFarland JG, Bougie DW. Drug-induced immune thrombocytopenia: pathogenesis, diagnosis, and management. Journal of Thrombosis and Haemostasis. 2009 Jun 1;7(6):911-8.

Round 2

Reviewer 3 Report

n/a

Author Response

Thank you for your revisions. The content was adjusted to meet the minimal word count. Further, the draft with reviewed by multiple members of our team, including native English speaking colleagues. Thank you kindly.